# The Heavenly Passage Known in the West as Reissner's Fiber

Lawrence Wile 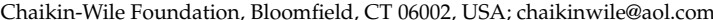

Chaikin-Wile Foundation, Bloomfield, CT 06002, USA; chaikinwile@aol.com

**Abstract:** This article explores the hypothesis that Reissner's fiber, an enigmatic, anomalous, thread-like structure that runs from the center of the brain to the end of the spinal cord, is the neural substrate of suprasensory perceptions of the divine. Justification for this hypothesis derives from a comparative study of descriptions of the "subtle body" from ancient esoteric traditions, testable speculations about altered states of consciousness correlated with the subtle dynamics of the fiber, and the fiber's evolutionary trajectory in humans from its perinatal involution to its potential regeneration. While adequate testing of the hypothesis will require new technologies, preliminary investigations are underway. The goal of this research is to promote research about Reissner's fiber with the hope that it could lead to the discovery of a universal religious experience underlying the transcendent unity of religions.

**Keywords:** neurotheology; comparative religion; esoteric traditions; quantum neurobiology; yoga; Taoism; Kabbalah; Reissner's fiber

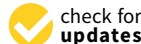



## 1. Introduction

Religious faith is problematic because the beliefs toward which it is directed are diverse and often incompatible, ambiguously justified and communicated, and ultimately beyond intellectual comprehension. The purported suprasensory perceptions of the divine by saints and prophets are ineffable. Confronted with these problems, most religious scholars have disallowed the quest to know what happens inside their brains. Only analyses of public communications and behaviors in the context of historical and cultural forces are legitimate (von Stuckrad 2003). Similarly, neuroscientists, psychologists and sociologists investigating religious experiences generally adopt "methodological agnosticism" or "methodological atheism" (Porpora 2006; Tolstaya and Bestebreurtje 2021). Religion has been interpreted as a manifestation of psychosis (Murray et al. 2012), a symptom of temporal lobe epilepsy (Devinsky and Lai 2008), an illusion created to fulfill infantile wishes, a neurotic defense against painful realities (Freud 1928, 1939), and a social construct created to gain power and promote social order (Restivo 2021). This article proposes that the truth or falsity of a perennialist interpretation of religion can be tested as a scientific hypothesis by investigating Reissner's fiber (RF), an anomalous, enigmatic thread-like structure that ensheathes the central axis of the central nervous system (Muñoz et al. 2019; Wile 2016) as the possible source of suprasensory perceptions of the divine.

Justification for this investigation derives from a comparative study of descriptions of the "subtle body" from ancient esoteric traditions, testable speculations about altered states of consciousness correlated with the subtle dynamics of the fiber, and the fiber's evolutionary trajectory in humans from its perinatal involution to its potential regeneration. Owing to the fiber's spectacularly strategic location and unique structure, it has the potential to radically alter the relationship between consciousness and reality and thereby provide new perspectives on the dialogue between science and religion. The goal of this article is to encourage research about RF with the hope that the politicized conflicts among various religions, and among secular humanists, atheists and religious believers, can be superseded by the sharing of a common neuropsychological ground.

## 2. Subtle Body

The "subtle body" lives in the contentious, mysterious borderland between science and religion, perennialism and constructivism, psychosis and transcendence, body and spirit, mind and matter, and science and pseudoscience. A comparative study of descriptions of the "subtle body" from various esoteric traditions leads to an identification of RF with its central axis. The following focuses on Hindu, Taoist and Jewish traditions because they trace their origins to prehistoric times and provide the clearest correspondences with RF. This identification brings other later descriptions, such as Apostle Paul's description of the "spiritual body" and the Sufi's description of the *Jism Latif*, closer to an anatomical correspondence. Amongst the diverse, incompatible phenomenologies, practices and doctrines of various esoteric traditions, RF could, literally, be the common thread.

RF was first identified with the central circuit of the "subtle anatomy" in 1939 by Theos Bernard (1940), a celebrity, scholar-practitioner of yoga and the first American ever initiated into tantric yoga practices by the highest Lama in Tibet. In his autobiographical *Heaven Lies Within Us*, he wrote "Inside this central (*Sushumna*) nadi, the Yogi identifies an invisible nadi known in the West as the fibre of Reissner, but which is known here as *Chittra* (the Heavenly Passage, in Sanskrit)". This identification is based on yoga's descriptions of concentric tubes (*nadis*) that conduct a female divine energy (*kundalini*) from the base of the spine to the brain to achieve higher states of consciousness. *Kundalini* is described as a sleeping serpent that lies coiled in a triangular-shaped region at the base of the spine called the *Mooladara chakra*. The innermost nadi, the *Brahma nadi*, is ensheathed by the *Chittra* and *Vajra nadis*, and travels through a passageway called the *Sushumna nadi*. The *Shatchakra-nirupana* ("Description of and Investigation into the Six Bodily Centres"), written in 1526 and translated from the Sanskrit in 1924 by Sir John Woodroffe (2017), a prominent British Orientalist whose writings helped overcome tantric yoga's lurid, exotic reputation, provides a detailed description of the *nadis* inside *Sushumna*: *Vajra, Chittra and Brahma*. The second verse states, "Inside the *Vajra* is *Chittra*. She is subtle as a spider's thread". The forty-eighth verse states that the *Brahma randhra*, the entrance to the *Brahma nadi* which is inside the *Chittra nadi*, is, "extremely subtle and like unto the ten-millionth part of the end of a hair". The *Sushumna nadi* corresponds to the central canal, a minute passageway running through the spinal cord. The *Chittra nadi* corresponds to the several-micron diameter fiber. The *Brahma randhra* corresponds to its hollow opening. RF coils in a triangular-shaped region at the base of the central canal called the terminal ventricle which is surrounded by secretory and sensory cells of unknown function (Motavkin and Bakhtinov 1990). While the central canal, RF, and the terminal ventricle were not observed by anatomists until the 19th century, they could have been "seen" by yogis with the sensory neurons (cerebrospinal fluid (CSF)-contacting neurons) that surround them.

While the hypothesis that CSF-contacting neurons can generate visual percepts is speculative, it is consistent with neuroscientific knowledge. The receptors surrounding the fiber contain photosensitive proteins (Blackshaw and Snyder 1999) and its dendrites project to a relay station along the visual pathway (Guillery and Sherman 2002). Mediative practices, which inhibit external sensory input (Mohandas 2008), could make RF shine like stars on a dark night.

Corresponding to the *nadis* of yoga are the meridians of acupuncture and esoteric Taoism, described in detail in *The Great Compendium of Acupuncture and Moxibustion* written during the Ming Dynasty (1368–1644). During the 1960s, Kim Bong-han, a North Korean scientist, investigated the anatomical basis of acupuncture (Bong-han 1962; De Vernejoul 1985). He injected radioactive phosphorous into an acupuncture point on a rabbit linked to the Governing Vessel Meridian, an energy conduit that corresponds to the central *nadis*. It is the first meridian to form after conception and forms the dorsal component of the "microcosmic orbit", through which female (*yin*) and male (*yang*) manifestations of a universal divine energy (*Qi*) circulate. Photomicrographs show that the injected phosphorous labelled RF, which Bong-han renamed as the "Neural Bonghan duct".

Bong-han's controversial discovery has recently been confirmed (Lee 2008; Soh 2004) and the ducts have been investigated as optical channels for coherent biophotons (Soh 2009). However, while the "microcosmic orbit" is a key element of esoteric Taoist practices, such as the coordinated movements, breathing and meditation of *Qigong,* the meditative inner transformations of *Neigong* and the meditative alchemy of *Neidan,* research on the Bonghan ducts has focused on its role in medical treatments.

Corresponding to the *nadis* of yoga, and the meridians of acupuncture and esoteric Taoism, are the paths of the *Sephirot*, alluded to in one of the earliest kabbalistic texts, *Sepher Yetzirah*. The *Sephirot* are emanations represented by 10 spheres symbolizing divine attributes. In the 16th century, Isaac Luria superimposed a complex, multidimensional, dynamic system of reconfigured *Sephirot,* wherein cosmic, male and female spiritual entities (*Partzufim*) interact to redeem the exiled female, divine presence (*Shekinah*). According to Lurianic Kabbalah, the *Sephirot* originated from the transtemporal, infinite *Ein-Sof.* Through a process of self-withdrawal (*Tzimtzum*) that began 15,340,500,000 years ago (Kaplan 1993), followed by a series of contractions, descents and concealments, the *Ein-Sof* emanated Adam Kadmon, the unmanifest potential of Adam Ha-Rishon who was created in the Garden of Eden. Next, Adam Kadmon emanated the 10 spheres of the *Sephirot.* First, they were configured as concentric circles (*Igullim*) and then a linear configuration (*Yosher*) consisting of three vertical pillars, the divine image in which man was created (Fine 2003).

Wary of anthropomorphic conceptions of God, kabbalists have generally used anatomical interpretations of the *Sephirot* analogically or metaphorically. Nevertheless, from a perennialist perspective, descriptions of the *Sephirot*, *nadis* and *chakras*, and meridians share a common, experiential, ancient origin. Corresponding to the central *nadis* and the "microcosmic orbit" is the central pillar of the *Sephirot*.

The above descriptions of the "subtle body" overlook a key element of its mediation of the relationship between the human and the divine, sexuality. Yogic (Samuel 2010), Taoist (Wile 1992), and Kabbalistic (Segol 2021) practices emphasize sacred sexuality that mirrors a cosmic sexual polarity. The sexual dimorphism of RF is unclear, but in addition to the main branch of the fiber from the center of the brain through the center of the spinal cord is another branch from the center of the brain across a slit-like cavity (third ventricle) to the hypothalamus. The site of the fiber's connection to the hypothalamus is the preoptic nucleus (Enami 1954), which contains the sexual dimorphic nucleus, a key neural substrate of sexual behavior (Hoffman and Swaab 1989).

Owing to the obscurity of the origins and the ambiguous nature of descriptions of the "subtle body", it is unclear whether they are fact or fiction. Peeling away layers of history using the methods of historical criticism, philology and archaeology cannot answer this question. But new developments in science and technology might.

### 3. Reissner's Fiber and the Divine

Because RF is a rare anomaly that undergoes rapid post-mortem degeneration and is too thin to be detected by current neuroimaging devices, correlations between consciousness and the fiber's activity are unknown. However, the altered states of consciousness that could be facilitated by the fiber have been investigated. RF is bathed in a variety of psychoactive substances. Among them are oxytocin (popularly known as the "love hormone") and endogenous cannabinoids, opioids, and psychedelics such as N, N-Dimethyltryptamine (DMT) which circulate in the CSF. Because RF binds similar substances (Caprile et al. 2003; Hess and Sterba 1973) and projects filaments to the CSF-contacting neurons that mediate their effects (Kohno 1969), it could transport them directly to nerve endings and produce more sustained and intense effects. Thus, RF could facilitate feelings of love, bliss, noesis and ego dissolution (Wile 2020), and, in the case of DMT, visions of angels and other supernatural phenomena (Strassman 2014) that are prevalent elements of religious experience.

Mystical experiences generated by RF's facilitation of the actions of endogenous psychoactive substances, though, would not justify faith in revelations, theophanies and prophecies from divine sources. Viewing RF's 5-nanometer diameter filaments, embedded

nanostructures and hollow core through the lens of modern physics, however, provides new perspectives on the borderland between the natural and supernatural. The mysteriousness, ineffability, immateriality and holism of the subatomic realm, and its intimate relationship with the forces of creation have inspired comparisons with the esoteric traditions (Afialo and Schipper 2012; Marin 2009; Restivo 1982; Scerri 1989)

Whether comparisons between modern physics and esoteric traditions point toward a common ontology or merely a shared mysteriousness is unclear. We are denied perception of the realm ambiguously reflected in the equations and experiments of physics because, somewhere, somehow, sometime along the pathway to conscious perception, it "collapses" into classical reality. This article proposes that direct, unmediated consciousness of "uncollapsed quanta" that comprise RF is the neural substrate of suprasensory perceptions of the divine.

Phenomenological analysis of the perceptions that might be correlated with the activity of RF will require regenerating the fiber using the tools of genetic and epigenetic engineering, perhaps supplemented with traditional spiritual practices such as yoga, acupuncture, and various methods of meditation. A neurophenomenological analysis will require measuring the fiber's activity, including its quantumness, with new technologies such as nanobots, artificial intelligence, brain–computer interfaces and new generations of neuroimaging devices with greater resolution and sensitivity.

Evaluating the truth or falsity of individual religious experiences that might be correlated with activity of RF is problematic. But, as communities of individuals endowed with RF emerge, an intersubjective consensual reality could replace rare subjective experiences and faith in mysteries from the past. Mass revelations would be possible.

Another variety of religious experience that might be evaluated by the categories of true and false are miracles. Volitional control of "uncollapsed quanta" might explain the purported paranormal powers that result from development of the "subtle body". These range from the *siddhis* of yoga such as levitation and psychokinesis to the miracles, transfiguration and resurrection of Jesus, to sowing the seeds of a new redemptive cosmogenesis. Based on recent measurements of collisions in the Large Hadron Collider, and the assumption of random quantum fluctuations, physicists predict that the universe is poised to undergo a new cosmogenesis (Carroll and Chen 2004). That event is so astronomically improbable that it is predicted to occur no sooner than $10^{58}$ years from now (Daley 2018). Volitional control of the quanta comprising RF could make that event imminent.

Volitional control of quanta has been the subject of speculation since the inception of quantum uncertainty. The most well-known instantiation of this idea was first proposed in 1935 by Sir Arthur Eddington (1935) and later developed in 1992 by Sir John Eccles and Friedrich Beck (Beck and Eccles 1992). They noted that the spatial quantum uncertainty of the vesicles that release neurotransmitters in synapses is about five nanometers, the thickness of the membrane surrounding them. By volitionally triggering electronic processes in synaptic vesicle membranes, the mind could control the brain.

The specificity of volitional control of physiological processes is remarkable. Individuals can control the firing of single motor neurons (Basmajian 1962) or differentially control the temperatures of adjacent fingers (Surwit et al. 1976). However, such control involves only single atoms that trigger a cascade of classical neural events. Harnessing volitional control of the quanta that comprise RF to achieve paranormal powers would require controlling the coherent, collective, cooperative behavior of innumerable quanta.

Until recently, such macroscopic quantum biological systems were thought to be impossible. However, newly discovered fractal, self-organizing, dissipative, and quantum chaotic levels of biological organization, recent observations of biological quantum coherences (Al-Khalili and McFadden 2015), and advances in quantum feedback and control using cavity quantum electrodynamic (cavity QED) systems have shown that the barriers to creating macroscopic quantum systems, "Schrodinger's cats", are "technical, not conceptual" (Wineland 2013). RF, suspended within the brain's ventricular cavities and central canal, is uniquely well suited to function as a biological analog of a cavity QED

system: photoreceptors with single photon sensitivity that surround the fiber could provide feedback of its biophoton emissions to the brain—the most complex information processing system in the known universe—which, in turn, could control the fiber's quantumness. Perhaps contemplative practices are conducive to the subconscious operation of such a neural quantum feedback and control system. Alternatively, a biofeedback system using neuroimaging devices might facilitate volitional quantum feedback and control. Mathematical models of the fiber can provide a path toward attuning it to subtler levels than those that might have been attained by the primordial mystics.

For the past 20 years, my team, led by Professor Vasili Kharchenko, Harvard-Smithsonian Center for Astrophysics and Harvard Physics Department, and Professor Alexander Sergienko, Department of Physics, Boston University Photonics Center, have searched for quantum coherences in RF using a novel correlation time-resolving infrared microspectroscope using a scanning confocal microscope, superconducting single-photon detectors and femtosecond pulsed lasers to measure photon emissions from RF in transparent zebrafish larvae. Thus far, the signal to noise ratio has been too low to draw meaningful conclusions. This year, I have begun a collaboration with Philip Kurian, Principal Investigator and Founding Director, Quantum Biology Laboratory, Howard University, to develop a basic atomistic model of RF, analyze cooperative quantum behavior in the simplified model of RF, and investigate the enhancement of such cooperative quantum behavior in the RF model due to integration of calcite and sulfur-containing amino acids that are associated with the fiber.

### 4. Neurohistory of Reissner's Fiber

According to Hindu, Taoist and Jewish traditions, a primal harmony between humanity, the cosmos and the divine shattered in the mists of prehistory and will be elevated to a higher harmony in the future. This perplexing history, which lies at the heart of theodicy, is consistent with the evolutionary neurohistory of RF.

During the evolutionary transition from apes to *homo sapiens*, RF began to involute perinatally (Muñoz et al. 2019). Humankind also took an evolutionary leap. They liberated themselves from biologically programmed behaviors by freely creating an expanding world of knowledge generated by the cognitive processing of feelings and sensations.

This evolutionary leap was propelled by the creation of what linguists call the digital infinity of language. Combinations of less than 30 sounds can form an infinite variety of words, which, although lacking any resemblance to the realities they represent, can communicate meaning to others. Because there can be no gradual transition from an analogue system of language to a digital one, Noam Chomsky hypothesized that the creation of the digital infinity of language occurred 100,000 years ago as the result of a chance mutation that produced the neuroanatomical basis of language in "near perfect form". It was "surprisingly perfect'—just as we might expect had it been designed by 'a divine architect'" (cited in Knight 2008).

The first step toward the creation of knowledge as an abstract creation of the mind was what Einstein called the "metaphysical original sin", the conceptualization of physical objects as existing independently from the subjective stream of sensations. It is a "sin" because there is no logical distinction between them. Einstein believed its commission is necessary, however, to avoid solipsism (cited in Schlipp 1998).

The positing of independently existing physical objects constitutes what Einstein called the primary layer of thinking. Knowledge progresses by creating increasingly unified layers. Paradoxically, the logical conclusion of the unification of knowledge derived from sensory input is an all-embracing infinity that is so remote from experience as to be nothing. Additionally, the ultimate unification of empirical knowledge bears the mark of the metaphysical original sin that created the duality between subject and object. The subject who observes the subject observing the subject observing the subject vanishes in an infinite regress.

Recorded religious history began 6000 years ago when knowledge progressed to its ultimate unity, known as Brahman, Tao and Ein-Sof by Hindu, Taoist and Jewish esoteric traditions, respectively. But religion is not based on self-referential paradoxes and infinite regresses. It is based on the direct experience of ultimate reality, atonement for the metaphysical original sin.

This direct experience of ultimate reality could be achieved by creating an infinity mirror of RF reflecting itself in the mirror of the stratified, interconnected, recursive web of language. When the unification of knowledge reached its logical conclusion, selective pressures had nearly eliminated the regulators of genetic expression that allowed the persistence of RF into adulthood. But there were rare exceptions. In those rare individuals whose fibers persisted into adulthood, the infinity mirror of RF, and Brahman, Tao and Ein-Sof raised the subconscious perception of RF's uncollapsed quanta to conscious, suprasensory perceptions. That explosive transformation of consciousness was the origin of religion. Paradoxically, it was end of humankind's primordial religious paradise.

That event was recorded by the revelations received by Hindu *rishis*, the Three Sovereigns and Five Emperors of China, and by Adam from the angel Raziel. Their knowledge was transmitted along an elite chain of initiates, from *rishis* who heard divine revelations to those who remembered them and developed them intellectually, from the Yellow Emperor to Lao Tze, and from Adam to Enoch, Noah, Joseph and Moses. The language used by that chain of initiates was constituted by divine energies that resonated with the essence of reality. Eventually it was translated into ordinary language (Holdrege 1996).

Religions that followed the purported primordial traditions of Hinduism, Taoism and Judaism, such as Buddhism, Zen Buddhism, Confucianism, and the Abrahamic religions such as Christianity and Islam, maintained continuity with their origins in spite of their theological differences. Jesus, for example, proclaimed, "For truly, I say to you, until heaven and earth pass away, not an iota, not a dot, will pass from the Law until all is accomplished". (Matthew 5:18), implying continuity with the esoteric tradition of deriving meaning from the geometry and vibrations of the Hebrew letters of the Torah. Significantly, Christian tradition says that grace presupposes nature. From this perspective, RF's role is to be expected because God's grace is inseparable from nature.

As human societies became increasingly governed by verbally communicated and externally enforced laws, the consciousness generated by RR become increasingly difficult to integrate with the consciousness of consensual reality. Individuals whose behaviors were influenced by RF were at a selective disadvantage. They would have been the first schizophrenics. Indeed, acute psychotic exacerbations in schizophrenics correlate with increased secretions of the subcommissural organ, the main source of RF (Vilkov et al. 1984).

The evolutionary persistence of schizophrenia presents a paradox. In spite of its obvious selective disadvantages, its prevalence in the population has remained constant and is currently increasing. If schizophrenics are the conflicted vestiges of shamans, saints and prophets, they might serve to evoke dormant, adaptively advantageous, religious experiences in normal individuals. If RF is the neural substrate of suprasensory perceptions of the divine, it now imprints its effects on the collective unconscious and on the perinatal matrices of the unconscious formed (Grof 1976) when RF guided our neurons in the womb.

For more than 100,000 years, selective pressures have favored the perinatal suppression of the genetic machinery that produces the fiber. The wisdom of esoteric traditions and the wisdom of natural selection are in conflict. Natural selection has determined that shutting down the genetic machinery that produces RF promotes survival and reproductive fitness. A neurocosmological interpretation of esoteric knowledge suggests that RF is the path to Enlightenment and cosmic redemption. Humankind stands at the crossroads.

## 5. Conclusions

Recently, two historians of neuroscience, Regis Olry and Duane Haines, dubbed RF the "Devil according [to] Baudelaire" whose "loveliest trick is to persuade you that he does not exist!" (Baudelaire 2017, p. 1). Given the fiber's unique, complex structure, its

770-million-year evolutionary conservation and its strategic location, neuroscience's neglect is puzzling. Perhaps, the identification of RF as the Devil according to Baudelaire is more than a playful analogy. In their article "Reissner's Fibre: The Exception Which Proves the Rule, or the Devil According [to] Charles Baudelaire?", Olry and Haines (2003) cryptically invite the reader to fill in the gap of the concluding ellipsis: "From time out of mind, the Devil was always called the Devil: now we do not even know if it exists, and even less what it might be used as". But, before "time out of mind", the Devil was known in Hebrew as *satan,* whose every action, according to the Talmud, is for the sake of Heaven. RF's hidden identity as the shattered fragments of the "subtle body" might lure scientists to peer beneath Satan's cape.

When Job protested against the afflictions Satan had convinced God to inflict on him, the Lord responded "Who is this that obscures my plans with words without knowledge? Where were you when I laid the earth's foundation? What is the way to the abode of light? Additionally, where does darkness reside?" (King James Bible [1769] 2017), implying that answers to these questions would lead to an understanding of the intellectually incomprehensible incompatibility between God's omnipresent, omniscient omnibenevolence and human suffering. Now, 2500 years after God posed these questions to Job, science has reached an impasse on the road toward answers. The regeneration of RF provides a new path toward transcending the limits of the intellect. Or, perhaps the material constituents of the fiber are a provisional scaffolding that can fall away and be replaced with a subtler, stable configuration of its immaterial constituents.

**Funding:** This research has no external funding.

**Institutional Review Board Statement:** Not applicable.

**Informed Consent Statement:** Not applicable.

**Data Availability Statement:** Not applicable.

**Conflicts of Interest:** The author declares no conflict of interest.

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
