# Peer review of "The Heavenly Passage Known in the West as Reissner’s Fiber"

_religions, doi:10.3390/rel13030248_

Round 1

Reviewer 1 Report

I feel ovewhelmed by this paper, which clearly goes beyond my expertise area, and hence there is little I can say.

In any case, I have some familiarity with neurological studies of religion, like those pionering of Newberg and D'Aquili, and later those of Mario Beauregard, who - to some extent - meant an end to former research and claimed that no a 'God spot' could be identified in the neural circuits.

For that reason, I have serious doubts that the present research could add some new pice to this mosaic, and still more to locate a more reliable area in the barin linked to religious functions or experience. The provided evidence is indeed scarce and most of the arguments are conjectural.

Author Response

Thank you for your comments. My paper differs from Newberg and D’Aquili’s and Mario Beauregard’s. They performed MRI investigations of meditating nuns. I propose regenerating Reissner’s fiber, investigating the resulting altered states of consciousness and correlating them with measurements of the activity of the fiber using new generation of neuroimaging devices.

The evidence is scarce and the arguments are conjectural. The goal of my paper to encourage future research about this neglected structure to gather new evidence

Reviewer 2 Report

The article is an interesting theory about the possibilities of connecting religion and science. The goal of the article seems good to me, as it aims to encourage research into the significance of the Reissner fiber in relation to religious experience.

I have two questions:

  1. How can we speak of finding a universal religious experience if we do not use the phenomenological method, which is the path to the discovery of this general experience? In the Philosophy of religion there many authors who, by using phenomenological method analyze general religious phenomena. Reissner's fiber is a material scientific fact, not a phenomenological phenomenon, which would be accessible to phenomenological analysis. How can a scientific confirmation of the existence of RF lead to the discovery of a universal religious experience without using the phenomenological method, which is used in the Philosophy of religion? The scientific method based on observations and technical instruments is quite different from the phenomenological method, used by the Philosophy of religion.
  2. Why does the author focuses only on esoteric traditions and omits, for example, the Christian tradition, although in the text he or she uses typical Christian concepts such as original sin, Heavens, redemption, or restoration of original harmony? Is there any special reason for this narrow selection? The Christian tradition, for example, says that grace presupposes nature (latin: Gratia supponit naturam), which means that God’s grace does not supply us with a new nature but works in the nature it finds. In this sense, from the point of view of Christian doctrine, Reissner's fiber is something expected, because God's revelation must have some material basis on which to anchor.

Author Response

Thank you for your comments. I completely agree that phenomenological methods are essential for the search for universal religious experience. I have therefore added material to highlight this need. Please see lines 166-180.

I focused on Hindu, Taoist and Jewish esoteric traditions because they trace their origins to prehistoric times. However, there is continuity between Christianity and those purported primordial traditions. Your reference to Christianity’s belief that grace presupposes nature is significant and I have added it to my paper. (see 292-301)

Reviewer 3 Report

It is a fascinating and thought-provoking study. I learned a lot from my reading and enjoyed the flow of your thoughts. 

You might consider redefining the basic goal of your argument, better clarifying how exactly it contributes to the evaluation of the truth value of religious events? it seems that your analysis is content-neutral and relates only to the authenticity of religious experience itself, not to any concrete claim.

I would also recommend adding a word of explanation for your choice to concentrate on Buddhism and (Lurianic) Kabbala. Is it a random choice or are they especially related to the described phenomenon?

in line 238 the Einstein quotation should read "metaphysical original sin".

Author Response

Thank you for your thoughtful and thought-provoking comments. Your criticism about the content neutrality of my analysis and the absence of concrete claims is valid. I, therefore added material to specifically address the problem. Please see lines 173-185.

I focused on Hindu, Taoist and Jewish esoteric traditions because they trace their origins to prehistoric times and provide the clearest correspondence to Reissner’s fiber. I added this explanation in lines 49-52.

Round 2

Reviewer 1 Report

I appreciate the effort the authors have made to address my scepticism regrading the submitted article.

I feel still it too esoteric for my perhaps to rational approach to the scientific study of religion.

Nevertheless, considering other opinions, I think the article can be published in its current form and help to open some discussion the suggested topic, which is clearly very original and out from the current trends in the study of religion